# BoardgameQA: A Dataset for Natural Language Reasoning with Contradictory Information

**Mehran Kazemi, Quan Yuan, Deepti Bhatia, Najoung Kim, Xin Xu,**
**Vaiva Imbrasaite, Deepak Ramachandran**
Google Research
{mehrankazemi, yquan, bhatiad, njkim, xxujasmine,
vimbrasaite, ramachandrand}@google.com

## Abstract

Automated reasoning with unstructured natural text is a key requirement for many potential applications of NLP and for developing robust AI systems. Recently, Language Models (LMs) have demonstrated complex reasoning capacities even without any finetuning. However, existing evaluation for automated reasoning assumes access to a consistent and coherent set of information over which models reason. When reasoning in the real-world, the available information is frequently inconsistent or contradictory, and therefore models need to be equipped with a strategy to resolve such conflicts when they arise. One widely-applicable way of resolving conflicts is to impose preferences over information sources (e.g., based on source credibility or information recency) and adopt the source with higher preference. In this paper, we formulate the problem of reasoning with contradictory information guided by preferences over sources as the classical problem of *defeasible reasoning*, and develop a dataset called BoardgameQA for measuring the reasoning capacity of LMs in this setting. BoardgameQA also incorporates reasoning with implicit background knowledge, to better reflect reasoning problems in downstream applications. We benchmark various LMs on BoardgameQA and the results reveal a significant gap in the reasoning capacity of state-of-the-art LMs on this problem, showing that reasoning with conflicting information does not surface out-of-the-box in LMs. While performance can be improved with finetuning, it nevertheless remains poor.

## 1 Introduction

A fundamental goal of AI since its early days has been automatically applying logical or deductive reasoning to draw new conclusions from existing knowledge [29, 20]. Since a large amount of knowledge is available in the form of natural language, tremendous effort has been put into developing models that can understand and reason over natural language [23, 42, 54, 32, 12, 57] (see [35] for a survey). Recent years have seen substantial improvements in this direction thanks to advancements in pretrained language models (LMs) [8, 9] that can handle unstructured data more flexibly, combined with advanced prompting techniques [52, 31], and modular reasoning approaches [23, 12].

Existing work in automated reasoning in natural language usually assumes that the provided knowledge is consistent and reliable. But in many applications, the collection of information one has to reason with is inconsistent and contradictory. This is the case, for instance, when reasoning is performed with information found in different online sources or social media (e.g., retrieval-augmented LMs [17, 3]). When input sources are contradictory, one can consider various strategies to resolve the contradictions. One simple and practical formulation, which we adopt in this work, is to resolve the conflicts based on preferences over the information sources: when a con-

flict arises, the information from the source with a higher preference should be used to solve the reasoning problem. Depending on the application, preferences can be assigned based on different criteria, e.g., based on the credibility of websites or social media users, or based on the recency of the information with newer information being preferred over older information. Exceptions to generics can also be expressed as preferences; for example, generic knowledge such as *"birds fly"* (see also [6]) should be overridden by exceptions such as *"penguins are birds but do not fly"* (see also [1]) when reasoning about penguins. Figure 1 demonstrates an example of a reasoning problem with conflicting information, where the conflict is resolved based on recency.

Reasoning with conflicting information guided by preferences can be formulated as a form of the classical *defeasible reasoning* problem [33, 19, 28]. In this work, we study the reasoning ability of LMs in this setting. Toward this goal, we create a synthetic dataset where each example contains a defeasible theory (a set of input facts, possibly contradictory rules, and preferences over the rules), and a question about that theory. Answering the questions in the dataset requires multi-hop reasoning and conflict resolution over the input theory. The

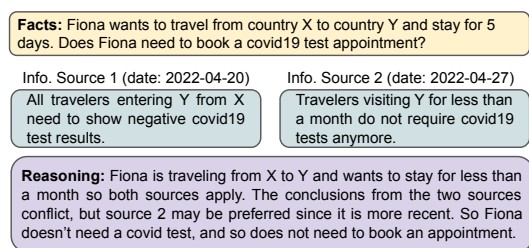

Figure 1: A reasoning problem with contradictory information (conflict resolved based on recency).

difficulty level (e.g., the depth, amount and type of conflicts, etc.) of the examples in the dataset can be controlled automatically, enabling targeted comparisons of various aspects of reasoning.

We also note that while a large number of logical reasoning benchmarks provide all the knowledge needed to answer questions [48, 41, 42, 18], such benchmarks do not reflect common real-world scenarios where implicit background knowledge plays an important role in reasoning. Moreover, models that translate the textual examples into logical form and then leverage off-the-shelf solvers may excel on these datasets, which does not reflect the true performance of such models in real-world applications. For these reasons, in BoardgameQA only part of the knowledge required to solve the problem is provided as input to the LM; the missing knowledge has to come from the LM itself.

The problems in our dataset are formulated as scenarios of a board game, hence we name it BoardgameQA[1]. A board game theme allows us to create synthetic scenarios with complex defeasible rules to reason about that seem natural when stated in text and hence allows background commonsense world knowledge to also be used. To the best of our knowledge, BoardgameQA is the first dataset for multi-hop reasoning *with contradictory inputs*. Figure 2 shows a sample example from the dataset where the conflict resolution and missing knowledge have been highlighted.

We benchmark various LMs on BoardgameQA and measure their defeasible reasoning capacity. Most notably, our results reveal that LMs perform poorly when reasoning with conflicting sources, especially in the few-shot setting (compared to the finetuning setting) suggesting that preference understanding and defeasible reasoning capacities do not surface out-of-the-box in pretrained LMs. Secondly, we find that smaller LMs perform poorly when not all of the required information is provided as input. These results highlight a critical gap in the reasoning capacity of current LMs, considering that reasoning over contradicting and incomplete sets of information is a common scenario in many applications, and is key for developing robust AI systems.

## 2 Related Work

Our work spans three dimensions: 1- text-based logical reasoning, 2- reasoning with conflicting sources, and 3- reasoning with incomplete information. In the following section, we briefly summarize the literature on each of these axes that relate to our work.

---

[1]All dataset variations used in our experiments can be found at: `https://storage.googleapis.com/gresearch/BoardgameQA/BoardgameQA.zip`. A variation with depth 4 (not used in our experiments) can also be found at `https://storage.googleapis.com/gresearch/BoardgameQA/depth4.zip`. All variations under `https://storage.googleapis.com/gresearch/BoardgameQA/` (including the ones mentioned above) are available under the *CC BY* license.

**Facts:** A few players are playing a boardgame. Here is the current state of the game. The dog has $3. The lion has $48. The frog has 81 dollars, and has a knife. [...]

**Rules:** R1: If the frog has more money than the lion and the dog combined, then the frog builds a power plant close to the green fields of the cat. R2: If something attacks the green fields of the cat, then it does not build a power plant near the green fields of the cat. [...]

**Question:** Does the frog build a power plant near the green fields of the cat?

**Proof:** We know the frog has $81, 81 is more than 3+48=51 which is the total money of the dog and lion combined, and according to R1 [...], and for the conflicting and higher priority rule R2 we cannot prove the antecedent "the frog attacks the green fields of the cat", so we can conclude "the frog builds a power plant close to the green fields of the cat". So the statement "the frog builds a power plant near the green fields of the cat" is proved and the answer is "yes".

Figure 2: A sample example from BoardgameQA that requires one hop of reasoning. The text in violet highlights conflict resolution and the text in blue highlights the missing information.

**Text-based logical reasoning approaches:** Earlier works on natural language logical reasoning have finetuned LMs to directly provide answers to logical reasoning questions [11, 4, 40, 18]. Later work showed that explicitly generating the entire proof leads to substantial improvements both in the case of finetuning and in the case of few-shot learning [31, 13, 58, 60]. In addition, modular reasoning approaches where the LM is used as a tool within a reasoning algorithm [23, 12, 51, 24] have been shown to achieve both performance gains and more precise intermediate proof chains. In this paper, we experiment with four types of approaches: 1- finetuning without explicit reasoning steps, 2- finetuning with explicit reasoning steps, 3- prompt-tuning with chain-of-thought (CoT) prompting [52], and 4- few-shot in-context learning with CoT.

**Text-based logical reasoning datasets:** Many datasets have been created to measure the logical reasoning ability of NLP models [48, 42, 61, 18, 45]. In Table 1, we provide a comparison of (a subset of) these datasets along three desired features in this work. All datasets compared contain only facts and rules that are non-contradicting. The datasets closest to our work are *CREPE* [30], FalseQA [21] and *ConditionalQA* [47]; the first two provide false pre-suppositions in the question which can be considered as statements that contradict the ground truth, and in last one the answers to the questions follow a *"If X then yes, if Y then no"* format.

**Reasoning with conflicts:** From the early days of AI, reasoning with conflicting information has been an important topic and many approaches have been developed to handle such conflicts [34, 33, 37]. The problem we study in this paper is an instance of defeasible reasoning [33, 19, 28] which has applications in various domains (especially in legal reasoning) [43, 16, 7] and has been argued to be one of the most important future directions in a recent survey of LM reasoning literature [56]. In defeasible reasoning, there are preferences over the rules and in the case of conflict between two rules, the conclusion from the higher preference rule is accepted. Previous work on defeasible reasoning with natural language has studied the problem of adjusting the probability of a conclusion based on new (single-hop) evidence [39, 27]. Our work extends this line of work by developing a dataset for multi-hop defeasible reasoning with preferences over sources.

**Reasoning with incomplete information:** Several existing reasoning benchmarks adopt a setup where part of the required information is missing and needs to come from the model itself [46, 5, 2, 50]. Some datasets also employ a setup in which none of the required rules are provided as input [49, 15, 45, 22]. Our work focuses mainly on cases where part of the knowledge needs to come from the model and another part of the knowledge is provided as input.

## 3   Background and Notation

We let $\mathcal{E} = \{e_1, \ldots, e_N\}$ and $\mathcal{P} = \{p_1, \ldots, p_M\}$ represent a set of entities and predicates. We represent a fact in the logical form using the triple notation $(e_i, p_j, e_k)$, where $e_i, e_k \in \mathcal{E}$ and $p_j \in \mathcal{P}$, and a rule as $r : r_b \to r_h$ where $r_b$ represents the body of the rule and $r_h$ represents the head. We use ! to indicate negation. A monotonic theory $\mathcal{T} = (\mathcal{F}, \mathcal{R})$ is a tuple containing a set $\mathcal{F}$ of (positive or negative) facts, and a set $\mathcal{R} = \{r_1, \ldots, r_{|\mathcal{R}|}\}$ of rules. We let $\mathcal{T} \vDash f$ represent that the fact $f$ can be derived from the theory $\mathcal{T}$ using the standard inference rules of logic (See Shoenfield [44]). For a monotonic theory $\mathcal{T} = (\mathcal{F}, \mathcal{R})$, if $\mathcal{T} \vDash f$, then for any theory $\mathcal{T}'$ such that $\mathcal{T}' = (\mathcal{F} \cup \mathcal{F}', \mathcal{R})$, we also have $\mathcal{T}' \vDash f$ (that is, adding new facts does not change previously derived facts).

**Defeasible Theory:** A defeasible theory $\mathcal{T}^{(d)} = (\mathcal{F}, \mathcal{R}, \mathcal{O})$ is a triple containing a set $\mathcal{F}$ of facts, a set $\mathcal{R} = \{r_1, \ldots, r_{|\mathcal{R}|}\}$ of rules, and a set $\mathcal{O} = \{r_{t_1} > r_{t_2}, \ldots, r_{t_3} > r_{t_4}\}$ of pair-wise relative

| Dataset → Feature ↓ | bAbI 15 | CLUTRR | FOLIO | Proof Writer | PrOntoQA OOD | AR-LSAT | ENWN | Leap of Thought | CREPE | False QA | Conditional QA | Boardgame QA |
|---|---|---|---|---|---|---|---|---|---|---|---|---|
| Contradictory Information | ✗ | ✗ | ✗ | ✗ | ✗ | ✗ | ✗ | ✗ | ∼ | ∼ | ∼ | ✓ |
| Incomplete Information | ✗ | ∼ | ✗ | ✗ | ✗ | ✗ | ✓ | ✓ | ✓ | ✓ | ✓ | ✓ |
| Auto. Diff. Control | ✓ | ✓ | ✗ | ✓ | ✓ | ✗ | ✗ | ∼ | ✗ | ✗ | ✗ | ✓ |

Table 1: A comparison of BoardgameQA with some of the widely-used logical reasoning datasets (bAbI 15 [53], CLUTRR [45], FOLIO [18], ProofWriter [48], PrOntoQA-OOD [42], AR-LSAT [61], ENWN [46], leap-of-thought [50], CREPE [30], FalseQA [21], and ConditionalQA [47]) in terms of three key features. We use ∼ in the case of incomplete information for CLUTRR because there is only a fixed set of information that needs to come from the model, in the case of automatic difficulty control for leap-of-thought because the depth of reasoning is fixed (difficulty is added through distractors), in the case of contradictory information for CREPE and FalseQA because the contradiction is with the ground truth not with another source of information, and in the case of contradictory information for ConditionalQA because while the answer to a question can be yes under one set of conditions and no under another set of conditions, the two sets of conditions are mutually exclusive.

priorities/preferences between rules.[2] The rules hold *defeasibly*, meaning the conclusion from a rule may be defeated by contrary evidence from a higher priority rule. This happens, for example, when one rule implies something is true but another rule with a higher priority implies it is false; in such cases, we accept the conclusion from the higher priority rule (see Figure 1). We let $\mathcal{T}^{(d)} \models f$ represent that $f$ can be derived from a defeasible theory $\mathcal{T}^{(d)}$ after resolving conflicts. Note that the initial facts $\mathcal{F}$ are internally consistent and always have priority over the derived facts. We assume the theory is *defeasibly consistent*, meaning whenever a conflict arises, the preferences can be used to resolve it. An example of a defeasible theory $\mathcal{T}^{(d)}$ is as follows:

**Example 3.1.** $\mathcal{F} = \{$*Tweety is a penguin.*$\}, \mathcal{R} = \{r_1 :$ *Penguins are birds.* $r_2 :$ *Birds fly.* $r_3 :$ *Penguins do not fly.*$\}, \mathcal{O} = \{r_3 > r_2\}$. *From the theory, one can first use $r_1$ to derive that "Tweety is a bird". Then, one can use $r_2$ to derive that "Tweety flies". However, one can also use $r_3$ to derive that "Tweety does not fly", which is in conflict with the previous derivations. Since $r_3 > r_2$, we accept the derivation that "Tweety does not fly".*

**Conflict types:** Conflicts can arise for rules whose heads cannot be simultaneously true, e.g., for two rules $r : r_b \to z$ and $r' : r'_b \to !z$. For a theory $\mathcal{T}^{(d)}$ with these two rules, $\mathcal{T}^{(d)} \models z$ in two cases: (a) $r$ has higher priority than $r'$ and we can prove $r_b$, and (b) $r$ has lower priority than $r'$ and we can prove $r_b$ but we cannot prove $r'_b$. In the first case, one does not need to take into account $r'_b$ for conflict resolution, but in the second case it is critical to take $r'_b$ into account. We name the first type of conflict *Type1* conflict and the second type *Type2*.

## 4 The BoardgameQA Dataset

We now describe how we create a dataset for measuring the ability of LMs in reasoning with conflicting inputs in a defeasible setup. Our dataset creation follows a backward story generation strategy similar to [55, 23]. Each example in the dataset contains a (defeasible) theory $\mathcal{T}^{(d)}$ and a question $q$. The goal is to predict whether $\mathcal{T}^{(d)} \models q$, or $\mathcal{T}^{(d)} \models !q$, or neither. Therefore, the label space for each question is $\{$*proved, disproved, unknown*$\}$. We next describe how we generate examples with the label *proved*; examples with the label *disproved* or *unknown* are created by modifying the examples with label *proved*.

The facts of each theory describe the current state of a board game, the rules of each theory represent the rules of the board game, and the questions are about the game. In the design of BoardgameQA, we include several variables that can be used to sample examples with varying levels of difficulty with respect to several finer-grained properties (e.g., depth, number and types of conflicts).

---

[2] Note: Many types of preferences can be converted into pair-wise relative preferences.

| Category | Description | Example Facts | Example Rule |
|---|---|---|---|
| Time Conversion | Compares the age of an entity to a certain age specified with different units. | The dog is 13 months and a half old | If the dog is more than a year old, then ... |
| Orthography | Asks about the letters in names. | The dog is named Paco. The cat is named Pashmak. | If the dog has a name that starts with the same letter as the name of the cat, then ... |
| Numeric Comparisons | Some numbers are required to be summed and then compared to other numbers. | The dog has two friends that are nice and five that are not | If the dog has less than 10 friends, then ... |
| Lexical Entailment | The fact and the rule body are not identical but the fact entails the rule body. | The dog assassinated the mayor | If the dog killed the mayor, then ... |
| World Knowledge | Some knowledge about the world is needed to connect the fact to the rule body. | The dog is currently in Montreal. | If the dog is currently in Canada, then ... |
| Event Times | Knowledge about times of events is needed to connect the fact to the rule body. | The dog is watching a movie that was released in 2005. | If the dog is watching a movie that was released after Covid19 started, then ... |
| Part Of | The fact and the rule body have a *part of* relation. | The dog is a nurse | If the dog works in healthcare, then ... |
| Affordance | The rule body is about a certain feature/affordance of the fact. | The dog has a knife | If the dog has a sharp object, then ... |
| Volumes | Knowledge of what objects fit in what other objects is required. | The dog has a ball with a radius of 15 inches. | If the dog has a ball that fits in a 28 x 35 x 35 inches, then ... |

Table 2: Categories, descriptions, and examples of incomplete information in BoardgameQA. For lexical entailment, world knowledge, event times, and affordance, a list of examples is written manually from which the sampling procedure can select. In others, examples are generated automatically.

**Entities and predicates:** We start with a predefined set of entities $\mathcal{E}$ (e.g., *dog, cat, lion,* etc.) and a predefined set of predicates $\mathcal{P}$ (e.g., *invite for dinner, attack the fields,* etc.) that we sample from to generate facts and rules. We use the animals as entities and the boardgame-inspired verbs/operations as our predicates. Using these entities and predicates, we can create facts such as *the dog attacks the fields of the lion*. To make the problem more challenging, we use different entities and predicates across training and test similar to [25]. The full list of entities and predicates is provided in Appendix C.3.

**Rule types:** We adopt a set of 6 rule templates containing existential and universal quantifiers, conjunctions, and missing information. The rules are as follows: 1- $\forall X : (X, p_1, e_1) \Rightarrow (X, p_2, e_2)$, 2- $\forall X : (X, p_1, e_1) \land (X, p_2, e_2) \Rightarrow (X, p_3, e_3)$, 3- $(e_1, p_1, e_2) \Rightarrow (e_2, p_2, e_3)$, 4- $(e_1, p_1, e_2) \land (e_3, p_2, e_2) \Rightarrow (e_2, p_3, e_4)$, 5- $(e_1, \hat{p}, \hat{e}) \Rightarrow (e_1, p_2, e_2)$, and 6- $\exists X (X, p_1, e_1) \Rightarrow (e_2, p_2, e_3)$, where $X$ represents a universally or existentially bounded variable, each $e_i$ represents an entity, and each $p_j$ represents a predicate. The fifth rule template corresponds to a rule where the predicate (or object entity) in the rule body may not be an element of $\mathcal{P}$ (resp. $\mathcal{E}$). For more information, see below.

---

**Algorithm 1** GenerateTheory

**Input:** Question $q$, Depth $d$

1: **if** d == 0 **then**
2:    *addToFacts*(q)
3: **else**
4:    $\mathcal{Q}, r = SampleRuleAndSubq(q)$.
5:    *addToRules*(r)
6:    **if** CoinFlip($p_{Conf}$) == Conflict **then**
7:      $\mathcal{Q}', r' = SampleRuleAndSubq(!q)$.
8:      *addToRules*(r')
9:      **if** CoinFlip($p_{ConfType1}$) == Type1 **then**
10:        $\mathcal{Q} = \mathcal{Q} + SubSample(\mathcal{Q}')$
11:        *addToPreferences*(r, r')
12:      **else**
13:        $\mathcal{Q} = \mathcal{Q} + RemoveOneSubquestion(\mathcal{Q}')$
14:        *addToPreferences*(r', r)
15:    **for** $q_i$ in $\mathcal{Q}$ **do**
16:      GenerateTheory($q_i$, d-1)

---

**Algorithm 2** SampleRuleAndSubq

**Input:** Question $q$

1: $r = SampleQuestion()$
2: **if** $r$ is a rule with incomplete info (type 5) **then**
3:    Sample $\mathcal{Q} = \{q_1, ..., q_n\}$ and $\hat{\mathcal{Q}} = \{q'_1, ..., q'_m\}$ s.t. $q$ can be derived from $\hat{\mathcal{Q}}$ and $r$, and $\hat{\mathcal{Q}}$ can be derived from $\mathcal{Q}$.
4: **else**
5:    Sample $\mathcal{Q} = \{q_1, ..., q_n\}$ s.t. $q$ can be derived from $\mathcal{Q}$ and $r$.
6: **return** $\mathcal{Q}, r$

**Selecting a question:** To generate each example, we first sample a question $q = (e_i, p_j, e_k)$ that should be proved or disproved, where $e_i$ and $e_k$ are sampled from $\mathcal{E}$ and $p_j$ is sampled from $\mathcal{P}$. We also sample the sign of the question (positive or negative). For example, we might sample the question *!(dog, attack the fields, lion)* asking whether *the dog does not attack the fields of the lion*. The question is then converted into natural language using a template (see Appendix C.3).

**Theory generation:** The theory generation is the main component of the dataset generation that constructs the facts, rules and question to be used in each example. A high-level description is provided in Algorithm 1 and an example generation is shown in Appendix C. We first sample some sub-questions $\mathcal{Q} = \{q_1, \ldots, q_n\}$ and a rule $r$ which has $\mathcal{Q}$ in its body and $q$ in its head, such that $q$ can be derived from $\mathcal{Q}$ and $r$. The sampling is done by first selecting one of the aforementioned rule types, then matching the head of the rule to the question $q$, and then sampling sub-questions $\mathcal{Q}$ based on the body of the rule. For example for the question *!(dog, attack the fields, lion)*, we might sample the first rule type (see the six types above), then $p_2$ will be mapped to *attack the fields* and $e_2$ will be mapped to *lion*, and we also sample a sub-question such as *(dog, unite with, cat)* and add the rule $\forall X : (X, \textit{unite with}, cat) \Rightarrow !(X, \textit{attacks the fields}, lion)$ to our set of rules. We then make a recursive call for each $q_i$ to generate new rules and facts for them.

We then decide whether a conflict should be introduced or not, by using a biased coin flip with $p_{Conf}$ representing the probability of conflict. If the decision is to produce conflicts, then we generate another set of sub-questions $\mathcal{Q}' = q_1', \ldots, q_m'$ and another rule $r'$ such that $!q$ can be derived from $\mathcal{Q}'$ and $r'$. Then we probabilistically decide if we want to generate a Type1 or a Type2 conflict using a biased coin flip with probability $p_{ConfType1}$. If the first case is selected, then $r > r'$ is added to the preferences. In this case, we can make recursive calls for all or a subset of the facts in $\mathcal{Q}'$. Otherwise, $r' > r$ is added to the preferences. In this case, we make recursive calls for *all but one* of the facts in $\mathcal{Q}'$ (selecting randomly) to ensure that $r'$ does not activate.

**Proofs:** We keep track of the facts, rules, and preferences during the generation process and turn them into proofs for the examples.

**Stopping criterion:** Every time we make a recursive call to the function in Algorithm 1, the example will contain one extra hop in its proof. We set the stopping criterion as the number of hops in the proof. Toward this goal, we included an argument $d$ in Algorithm 1 which corresponds to the target maximum number of hops in the proof; $d$ decreases by one every time we make a recursive call. When the algorithm is called with $d = 0$, instead of generating rules and sub-questions for the input question $q$, we simply add $q$ to our set of facts.

**Incomplete information:** We generate examples with incomplete information where part of the knowledge should come from the LM (corresponds to rule type 5). For a question $q$ in the theory generation phase, we sample sub-questions $\mathcal{Q}$ and rule $r$ such that $\hat{\mathcal{Q}}$ can be derived based on $\mathcal{Q}$ and $q$ can be derived from $\hat{\mathcal{Q}}$ and $r$. We then hide $\hat{\mathcal{Q}}$ from the model so the model has to derive it itself. Algorithm 2 describes the procedure. We use a separate body of world knowledge, commonsense knowledge, mathematical, and orthography reasoning for generating $\mathcal{Q}$ and $\hat{\mathcal{Q}}$ (see Table 2 for a high-level description and Appendix C.2 for more details). For example, for the goal *"the dog unites with the cat"* we generate the sub-question *"The dog is in Montreal."* and the rule *"If the dog is in Canada, then the dog unites with the cat."*. Then, an extra reasoning step is needed from the model to recognize that Montreal is in Canada.

We generate sub-questions and rules that require extra knowledge and reasoning with probability $p_{MissInfo}$; otherwise, we create sub-questions and rules that require no extra knowledge and reasoning. To make the problem more challenging, we only include some categories of extra knowledge and reasoning in the training set; this ensures that the models cannot simply learn the extra knowledge from the training set and use it in the test set.

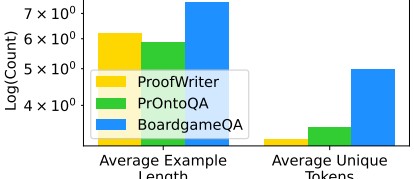

Figure 3: A comparison of BoardgameQA with ProofWriter [48] and PrOntoQA [41] in terms of average length of examples and average number of unique tokens per example on depth 3 of the datasets.

**Conversion to natural language:** Finally, once we generate the facts, rules, preferences, and question, we use manually constructed templates to turn each of them into a textual format. To make the problem more challenging, we use multiple templates per rule type and use some of the templates only in the test set (see Appendix C.3 for details). A comparison of BoardgameQA with other prominent deductive reasoning datasets in terms of the average length of examples and the average number of unique tokens per example is provided in Figure 3.

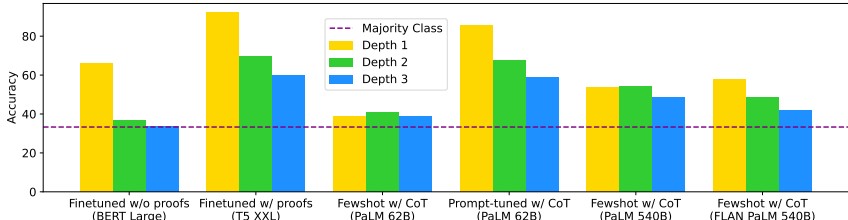

Figure 4: The model performances on depths 1–3 of the BoardgameQA dataset. Many models struggle on this dataset, especially with higher depths.

**Disproved and unknown examples:** So far, we described how to generate examples with the label *proved*. Generating examples with the label *disproved* can be done simply by first generating an example with the label *proved* and then negating the question. Also, generating examples with the label *unknown* can be done by perturbing the theory until the statement in the question cannot be derived from the theory (e.g., reducing the amount of money of the frog to 50 dollars in the example of Figure 2). We randomly select and apply the following perturbations to the theory and run a defeasible solver implemented based on the scalable solver in [28] on the resulting theory until the label becomes unknown: 1- change the predicate of a fact or a rule, 2- change the sign of a fact or an element of the rule, 3- replace a fact with a new fact, and 4- flip the order of a preference.

## 5    Experiments

One of the primary goals of our experiments is to verify if LMs are capable of reasoning in a defeasible setup. For this reason, we conduct experiments with various LM architectures (encoder-only, encoder-decoder, and decoder-only) and various pre-training and learning paradigms (finetune with and without proofs, prompt tuning, few-shot in-context learning, and instruction-tuned). Specifically, we test 1) finetuning BERT-large [14] with a classification head to predict the label directly, 2) finetuning T5 1.1 XXL [36] to generate the entire proof and then the label, 3) few-shotting PaLM 62B and PaLM 540B [9] where we provide demonstration examples and chain-of-thought (CoT) in the prompt (the CoT corresponds to the proof), 4) few-shotting the instruction-finetuned FLAN-PaLM 540B [10] with CoT, and 5) soft prompt-tuning [26] PaLM 62B with CoT where instead of providing a static prompt, we make the prompt embedding learnable and tune its parameters using the training data (the rest of the LM parameters are frozen). We report classification accuracy as the metric. We also report the *majority class* baseline (∼33% since our labels are balanced).

**Dataset sizes:** To gain a more detailed understanding of the models' defeasible reasoning capacity, we create several variations of BoardgameQA. The nature of the variation will be discussed in the remainder of this section with each experiment. For each variation, we sample 1000 examples for train, 500 for validation, and 1000 for test. We sample an equal number of examples from each label.

### 5.1    Can LMs Reason with Contradictory Inputs?

As explained in Section 4, BoardgameQA makes use of a number of variables that control various aspects of the dataset such as the amount and types of conflict and the amount of extra knowledge required. We start by creating a default version of the dataset that exhibits each of these properties to some degree by setting $p_{Conf} = 0.5$, $p_{ConfType1} = 0.5$, and $p_{MissInfo} = 0.5$. We then generate three datasets with depth 1–3 (i.e., requiring 1–3 hop(s) of reasoning, respectively), and measure the performance of our baselines on these datasets.

The results are in Figure 4. The tuned models perform reasonably on depth 1, but their performance substantially degrades on depths 2–3. This contrasts with previous observations for monotonic reasoning (e.g., in [11, 48]) where finetuned LMs reach near-perfect performance even on higher depths. This indicates that reasoning with contradictory inputs is more difficult even with finetuning. Moreover, we see that the few-shot models perform poorly across all depths showing that conflict resolution is not achieved out-of-the-box with pretrained models. This includes both PaLM and instruction-finetuned FLAN PaLM models. PaLM 540B performs better than PaLM 62B showing that

larger models may have higher capacity for defeasible reasoning. More insights from full confusion matrices can be found in Appendix A.

Hereafter, due to inference costs, we only experiment with finetuned BERT and T5, prompt-tuned PaLM 62B, and few-shot PaLM 540B, and with examples of depth 2 to keep a medium level of difficulty in terms of reasoning hops and enable measuring the effect of the other factors.

## 5.2 Does Correct Label Prediction Mean Correct Proof?

Recently, it has been shown that although large LMs achieve high accuracy on label prediction for (monotonic) reasoning task, they do so by generating spurious proofs that do not represent valid steps of reasoning [23]. There is also evidence that LMs frequently exploit spurious correlations in the data distribution to achieve high label accuracy, rather than reasoning purely deductively [59]. Hence we design evaluation metrics to reflect a more rigorous measure of accurate defeasible reasoning. In the case where a model predicts the label correctly, and the label is one of *proved* or *disproved* (where an actual proof exists), we measure whether the proof generated by the model is correct or not. For this purpose, we compute two automated proof accuracy metrics (named *Rule F1* and *Conflict F1*) and one manual metric (named *Overall Proof Accuracy*) as described below. For *Rule F1*, we extract the rules used in the golden proof and the ones in the proof generated by the model that are used to derive new facts (and ultimately, the goal). Then we compute the F1-score of the overlap of the two sets. For *Conflict F1*, we extract the conflict resolutions (corresponding to pairs of rules) used in the gold proof and the ones in the proof generated by the model, and compute the F1-score of their overlap. For *Overall Proof Accuracy*, we manually verify whether the proof is correct for $50$ sampled examples per model. We compute these metrics on depth 2 of the dataset.

According to the results in Figure 5, all models perform relatively well in selecting the correct set of rules for the proof. The few-shot model performs poorly on conflict resolution whereas the tuned models perform substantially better, suggesting that preference understanding and conflict resolution do not surface with simple few-shot prompting, and tuning is required for models to exhibit this capacity. Second, the models often generate wrong proofs, even when they predict the label correctly. The issue is less severe in the case of the prompt-tuned model but becomes more severe

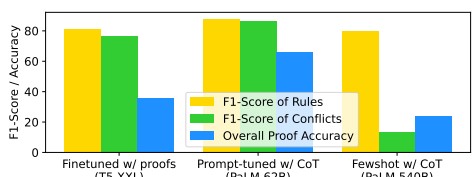

Figure 5: Proof accuracy metrics for various models on depth 2 of the dataset, when the label is predicted correctly.

for the finetuned and few-shot models. We provide examples of proof failures in Appendix A.

## 5.3 Do Conflicts Make Reasoning More Difficult?

We create four versions of BoardgameQA named NoConflict, LowConflict, Medium-Conflict, and HighConflict, with $p_{Conf}$ set to 0.0, 0.2, 0.5 and 0.8 respectively; other factors are kept the same. Note that the MediumConflict corresponds to the dataset in Figure 4. The results of the models on these datasets are reported in Figure 6. The performance of all models monotonically degrades as the number of conflicts increases, showing that conflict resolution is indeed a major factor in the difficulty of the problems. For

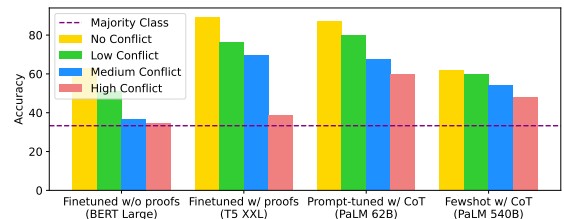

Figure 6: The model performances on four versions of the BoardgameQA dataset with various amounts of conflicts in them.

example, BERT performs above-random for the NoConflict and LowConflict cases, but the model performance drops to near-random on MediumConflict and HighConflict cases.

## 5.4 Which Conflict Type is More Difficult to Resolve?

To test which type of conflict (See sec. 4) is more difficult for the models, we create three versions of the dataset with varying proportions of Type1 vs Type2 conflicts, by setting $p_{ConfType1}$ to 0.2, 0.5, and 0.8 respectively. The first dataset mostly contains conflicts of Type1, the second contains both

conflicts in a similar amount, and the third dataset contains mostly Type2 conflicts. The other factors are kept constant across the datasets.

The results of the models are reported in Figure 7. We see that models perform slightly better on the dataset with mostly Type1 conflicts. This discrepancy between performance on Type1 and Type2 conflicts is intuitive because in the case of Type1 conflicts, the model can ignore the conflicting rule and whether its body can be proved, but in the case of Type2 conflicts, the model has to show that at least one of the elements in the body of the conflicting rule cannot be proved. In the case of tuned models, we furthermore observe that biasing the dataset toward one conflict type results in better performance overall. This might be because the model mostly needs to learn to resolve one type of conflict which may be easier than learning both.

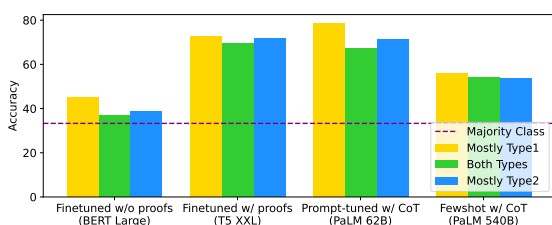

Figure 7: The model performances on three versions of the BoardgameQA dataset with different distributions on the type of conflicts.

## 5.5 Does Information Incompleteness Make Reasoning More Difficult?

As described in Section 4, we can control the amount of information incompleteness using a parameter which we named $p_{MissInfo}$. To test how the information incompleteness affects the performance of various models, we create three versions of our dataset with $p_{MissInfo}$ set to $0.2$, $0.5$ and $0.8$, which we name *KnowledgeLight*, *KnowledgeMedium* and *KnowledgeHeavy*, respectively.

The results are reported in Figure 8. We observe that as the amount of required knowledge increases, the performance of the fine-tuned models decreases accordingly. However, the performance of the prompt-tuned and few-shot models remain relatively unchanged, likely due to the larger size of the model and the extra amount of knowledge that is present in the model, as well as the fact that working with real-world knowledge might be easier for these models than with artificial knowledge.

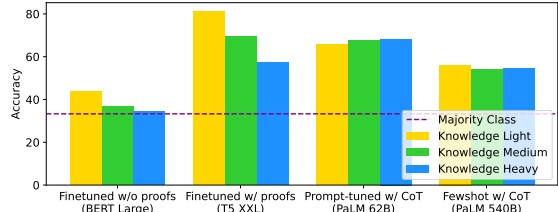

Figure 8: The model performances on three versions of BoardgameQA with various degrees of incomplete information.

## 5.6 Do Distractors Make Reasoning More Difficult?

We also measure the effect of distracting facts and rules on model performance. A distracting fact or rule is one that does not appear in the proof and does not change the label. In Figure 2, for example, *"the frog has a knife"* is a distracting fact. To this end, each time we call Algorithm 1, besides the sampled sub-questions, we also sample some distracting sub-questions and add them to the set of sub-questions. We create three versions of the BoardgameQA dataset where we add 0, 1, and 2 distracting facts in each step, which we name *NoDistractors*, *SomeDistractors*, and *ManyDistractors*, respectively.

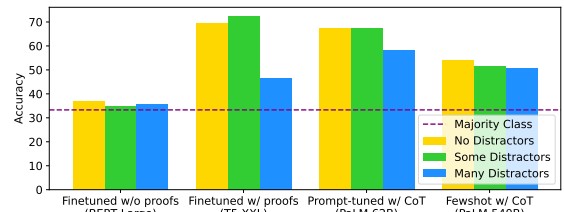

Figure 9: The model performances on three versions of BoardgameQA with various amounts of distracting facts and rules.

According to the results in Figure 9, the performance of the tuned models does not substantially degrade with a small number of distractors, potentially because the distractors can help the model avoid learning spurious correlations. However, their performance drops substantially with more distractors. Also, with more distractors, the performance of the few-shot model decreases monotonically,

although only marginally (this observation is consistent with the results of [42]). This shows that distractors (that are typically common in real applications) can also compound the problem difficulty.

## 6  Limitations

Our dataset, in its current form, focuses primarily on deductive logical entailment, where the problem is a classification problem ($label \in \{proved, disproved, unknown\}$), and the contradictions are also binary (i.e. one rule suggesting something is True and the other suggesting it is False). Future work can extend BoardgameQA and the analysis provided in this work to non-classification cases where 1- one needs to apply defeasible logical reasoning to answer questions such as ``Who will be attacked by the dog?'', 2- one needs to resolve non-binary conflicts where, e.g., one rule suggests ``the dog is currently in Canada'' and the other suggests ``the dog is currently in Australia'', 3- there are conflicts and preferences over facts as well, e.g., `Fact1: Fiona has travelled to every country in Europe, Fact2: Fiona has not travelled to the Scandinavian countries, Fact2 is preferred over Fact1`.

The current work assumes the initial state (facts) and the rules of the game are small enough to be included in the prompt. It is also limited to deductive reasoning with the *modus ponens* rule. Future work can extend BoardgameQA and our analyses to the cases where not all the facts and rules can be included in the prompt due to the limitation in the prompt length, as well as the case with other types of rules such as proof by contradiction, disjunction elimination, etc (see [42]).

In this work, we only studied one simple but highly practical solution to conflict resolution (i.e. based on preferences). Future work can extend BoardgameQA and the analysis in this paper to other natural types of conflict resolution. Note that in some applications, preferences for conflict resolution have to be assigned with great care and diligence to avoid unfair treatment of information sources.

## 7  Conclusion

In this work, we introduced BoardgameQA, a dataset for measuring the natural language reasoning ability of language models (LMs) in the presence of conflicting input sources. Our dataset furthermore includes scenarios in which the knowledge required for reasoning is only partially provided as input and additional information needs to come from the model itself. We tested several types of LMs on different variations of the dataset and observed that LMs perform poorly when reasoning with conflicting inputs. In the case of smaller models, the performance was also poor when additional knowledge from the LM is needed. Since reasoning over contradicting and incomplete sets of information is a common scenario in real-world applications, our results highlight an important gap in the reasoning capacity of current LMs. We hope our dataset can guide future work developing methodology to improve the reasoning ability of LMs under this setup, or finding alternative formulations of conflict resolution that better facilitate LM reasoning.

## Acknowledgements

We thank Yue Liu, Tania Bedrax-Weiss, and the anonymous reviewers for their valuable feedback.

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

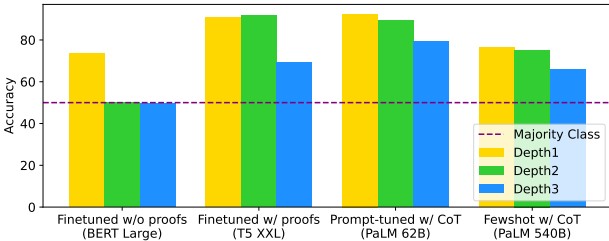

Figure 10: The model performances on various depths of a binary version of the BoardgameQA dataset.

**Depth 1 Example:** A few players are playing a boardgame. The current state of the game is as follows. The bee has 9 dollars. The dragon leaves the houses occupied by the liger. The shark has 48 dollars. The vampire is currently in Cape Town. And the rules of the game are as follows. Rule1: Regarding the vampire, if it has more money than the shark and the bee combined, then we can conclude that it disarms the bulldog. Rule2: There exists an animal which leaves the houses that are occupied by the liger? Then, the vampire definitely does not disarm the bulldog. Rule3: Regarding the vampire, if it is in France at the moment, then we can conclude that it disarms the bulldog. Rule1 is preferred over Rule2. Rule3 is preferred over Rule2. Based on the game state and the rules and preferences, does the vampire disarm the bulldog?

**Model Output:** Since the vampire has more money than the shark and the bee combined and according to Rule1 if the vampire has more money than the shark and the bee combined then we can conclude that it disarms the bulldog and for the conflicting and lower priority rule Rule2 we cannot prove the antecedent there exists an animal which leaves the houses that are occupied by the liger so the vampire disarms the bulldog. So the statement the vampire disarms the bulldog is proved and the answer is "yes".

**Error Analysis:**
- The model assumes *the vampire has more money than the shark and the bee combined*, but the amount of money the vampire has is not specified.
- Since Rule1 has higher priority than Rule2, if the model believes Rule1 activates, there should be no need to show that Rule2 does not activate.
- For Rule2, the model assumes that we cannot prove that *there exists an animal which leaves the houses that are occupied by the liger*, whereas we know the dragon does this based on the facts.

Figure 11: Highlighting some of the model errors for a depth 1 example when using PaLM 540B.

# A    More Experimental Results and Analysis

**Binary Classification:** Kazemi et al. [23] observed that reasoning with *unknown* labels is particularly challenging for few-shot LMs, because providing a natural chain-of-thought for *unknown* is difficult. To measure if the poor performance is merely due to the existence of examples with *unknown* label or due to conflict resolution being difficult for these models, we also created a binary version of the dataset for depths 1, 2, and 3 where only examples with *proved* and *disproved* labels are included.[3] The results are reported in Figure 10. We overall see similar patterns as the binary case, except for some improvements for the T5 model on depth 2.

**Examples of Model Failures:** To showcase some of the reasons why models struggle with the BoardgameQA dataset, in Figure 11  13 we show some example proofs where the model made a variaty of errors and in Figures 15–24 we provide specific examples where the model generated wrong proofs. Some of the dominant error cases (showcased in the examples) include: 1- hallucinating or misunderstanding conflicts and preferences, 2- not being able to correctly fill in the incomplete information, 3- misunderstanding logical rules, 4- failing to prove both elements in a conjunction, 5- getting distracted by distracting facts and rules and going on a wrong proof path, and 6- being unfaithful to the provided facts and rules and changing them so a proof can be found in the case where no proof exists.

**Confusion Matrices:** The confusion matrices for the model predictions with respect to the golden labels on the BoardgameQA dataset is provided in Figure 14 for various depths. One interesting observation is that for models tuned with proofs, while the model perform well at predicting unknown labels in lower depths, in higher depths they tend to generate proofs (with proved or disproved labels) even when the label is unknown (e.g., for examples with label unknown in depth 1 the prompt-tuned model predicts unknown for 268 instances, while in depth 2 it predicts unknown only for 2 instances). This may be because when the search space for a proof increases, LMs cannot verify all possible solutions and decide that the label is unknown. Instead, they start a path in the hopes that it ends in a proof.

**Standard Deviations:** Due to the high cost of experimenting with large language models, the results in the paper are reported based on a single run. To gain insight into the standard deviation of the

---

[3]In this case, we set $p_{Conf} = 1.0$ for the first call we make to Algorithm 1; Otherwise, the dataset will have a spurious correlation that can be exploited without doing any reasoning.

**Depth 2 Example:** A few players are playing a boardgame. The current state of the game is as follows. The gorilla is watching a movie from 1952. The woodpecker has a football with a radius of 28 inches, and swears to the duck. The zebra swims in the pool next to the house of the butterfly. And the rules of the game are as follows. Rule1: The gorilla will not enjoy the company of the vampire if it (the gorilla) works in healthcare. Rule2: The gorilla enjoys the company of the vampire whenever at least one animal swims inside the pool located besides the house of the butterfly. Rule3: If there is evidence that one animal, no matter which one, acquires a photograph of the peafowl, then the woodpecker calls the vampire undoubtedly. Rule4: Here is an important piece of information about the woodpecker: if it has a football that fits in a 58.4 x 61.5 x 59.6 inches box then it does not call the vampire for sure. Rule5: If at least one animal leaves the houses that are occupied by the dragon, then the vampire does not disarm the bulldog. Rule6: Here is an important piece of information about the gorilla: if it is watching a movie that was released after the first man landed on moon then it does not enjoy the companionship of the vampire for sure. Rule7: If something swears to the duck, then it leaves the houses that are occupied by the dragon, too. Rule1 is preferred over Rule2. Rule3 is preferred over Rule4. Rule6 is preferred over Rule2. Based on the game state and the rules and preferences, does the vampire disarm the bulldog?

**Model Output:** Since the woodpecker has a football with a radius of 28 inches and according to Rule4 if the woodpecker has a football that fits in a 58.4 x 61.5 x 59.6 inches box then it does not call the vampire so the woodpecker does not call the vampire. Since the woodpecker does not call the vampire and according to Rule3 if there is evidence that one animal, no matter which one, acquires a photograph of the peafowl then the woodpecker calls the vampire so the woodpecker does not acquire a photograph of the peafowl. Since the woodpecker does not acquire a photograph of the peafowl and according to Rule5 if at least one animal leaves the houses that are occupied by the dragon then the vampire does not disarm the bulldog so the vampire disarms the bulldog. So the statement the vampire disarms the bulldog is proved and the answer is "yes".

**Error Analysis:**
- Since Rule4 activates, the model assumes *the woodpecker does not call the vampire* without checking whether Rule3 (the higher priority rule) activates or not. Luckily for the model, though, Rule3 does not activate.
- While Rule4 indeed activates, the model makes a planning error by starting from Rule4. The correct proof plan is to start with Rule7 followed by Rule5.
- An initially wrong proof plan leads to consequent wrong steps in the proof planning (Rule4 is followed by Rule3).
- The model misunderstands the antecedent and the consequent of Rule3.
- The model jumps to Rule5 and arrives at a conclusion about the question, without proving the antecedent of Rule5.
- The model misunderstands the sign of the consequent of Rule5.

Figure 12: Highlighting some of the model errors for a depth 2 example when using PaLM 540B.

**Depth 3 Example:** A few players are playing a boardgame. The current state of the game is as follows. The butterfly swears to the wolf. The cougar has 8 friends, and is named Chickpea. The crow is named Charlie. The dove has a basketball with a diameter of 15 inches, and is named Tessa. The dove has a card that is white in color. The monkey has a card that is white in color. The monkey is named Mojo. The shark is named Paco. The worm is named Tango. The husky does not tear down the castle that belongs to the finch. And the rules of the game are as follows. Rule1: From observing that an animal brings an oil tank for the bison, one can conclude the following: that animal does not smile at the beetle. Rule2: If the frog refuses to help the dalmatian, then the dalmatian is not going to invest in the company owned by the beetle. Rule3: Are you certain that one of the animals manages to convince the ostrich and also at the same time smiles at the beetle? Then you can also be certain that the same animal negotiates a deal with the badger. Rule4: If the monkey has a card whose color appears in the flag of Netherlands, then the monkey does not hide the cards that she has from the pelikan. Rule5: The pelikan unquestionably shouts at the badger, in the case where the monkey does not hide her cards from the pelikan. Rule6: If the monkey has a name whose first letter is the same as the first letter of the shark's name, then the monkey does not hide her cards from the pelikan. Rule7: Here is an important piece of information about the cougar: if it has a name whose first letter is the same as the first letter of the crow's name then it manages to convince the ostrich for sure. Rule8: Regarding the dove, if it has a name whose first letter is the same as the first letter of the worm's name, then we can conclude that it enjoys the companionship of the dalmatian. Rule9: If the cougar negotiates a deal with the badger and the pelikan shouts at the badger, then the badger falls on a square of the mouse. Rule10: One of the rules of the game is that if the dove enjoys the companionship of the dalmatian, then the dalmatian will, without hesitation, invest in the company whose owner is the beetle. Rule11: The cougar will smile at the beetle if it (the cougar) has more than seven friends. Rule12: This is a basic rule: if the finch negotiates a deal with the pelikan, then the conclusion that the pelikan will not shout at the badger follows immediately and effectively. Rule13: This is a basic rule: if the husky does not tear down the castle that belongs to the finch, then the conclusion that the finch negotiates a deal with the pelikan follows immediately and effectively. Rule1 is preferred over Rule11. Rule2 is preferred over Rule10. Rule5 is preferred over Rule12. Based on the game state and the rules and preferences, does the badger fall on a square of the mouse?

**Model Output:** Since the cougar has 8 friends and according to Rule11 if the cougar has more than seven friends then the cougar will smile at the beetle so the cougar smiles at the beetle. Since the cougar smiles at the beetle and according to Rule2 if the cougar smiles at the beetle then the cougar does not negotiate a deal with the badger so the cougar does not negotiate a deal with the badger. Since the cougar does not negotiate a deal with the badger and according to Rule9 if the cougar negotiates a deal with the badger and the pelikan shouts at the badger then the badger falls on a square of the mouse so the badger does not fall on a square of the mouse. So the statement the badger falls on a square of the mouse is disproved and the answer is "no".

**Error Analysis:**
- Since Rule11 activates, the model assumes *the cougar will smile at the beetle* without checking whether Rule1 (the higher priority rule) activates or not. Luckily for the model, though, Rule1 does not activate.
- The model hallucinates a non-existing rule instead of Rule2 which makes it shortcut to a conclusion.
- The model fails to understand conjunction and assumes Rule9 activates by only showing the first antecedent activates.
- The model misunderstands the sign of the consequent of Rule9.

Figure 13: Highlighting some of the model errors for a depth 3 example when using PaLM 540B.

results, we trained the BERT model (our cheapest baseline) 10 times on each of the datasets from the main text and computed the standard deviations. The standard deviations ranged from 0.3 to 2.6.

# B  Experimental Details

We conducted our experiments on v3 TPUs for all the models, except for the 540B models where we used v4 TPUs due to their larger size. All the experiments were done using the T5X framework [38] available at https://github.com/google-research/t5x.

For the fewshot experiments, to ensure the demonstrations and the question fit in the prompt, due to the large size of the examples in BoardgameQA, we only included one example per label as demonstration (i.e. 3 examples one with label proved, one with disproved, and one with unknown in the case of 3-way classification datasets). For each example in the test set, we selected the demonstrations randomly from the training set, while ensuring that they contain both types of conflict resolution. For the prompt-tuning experiments, we used a prompt-size of 100 in all experiments, as it worked best in the experiments of [26]. We allowed a maximum of 50K training steps with a batch size of 8, a learning rate of 0.1, and a weight decay rate of 0.0001. We evaluated the model after each 1000 steps on the validation set and selected the checkpoint with the best validation accuracy for testing. For the finetuning experiments, we also set the batch size to 8 but set the learning rate to

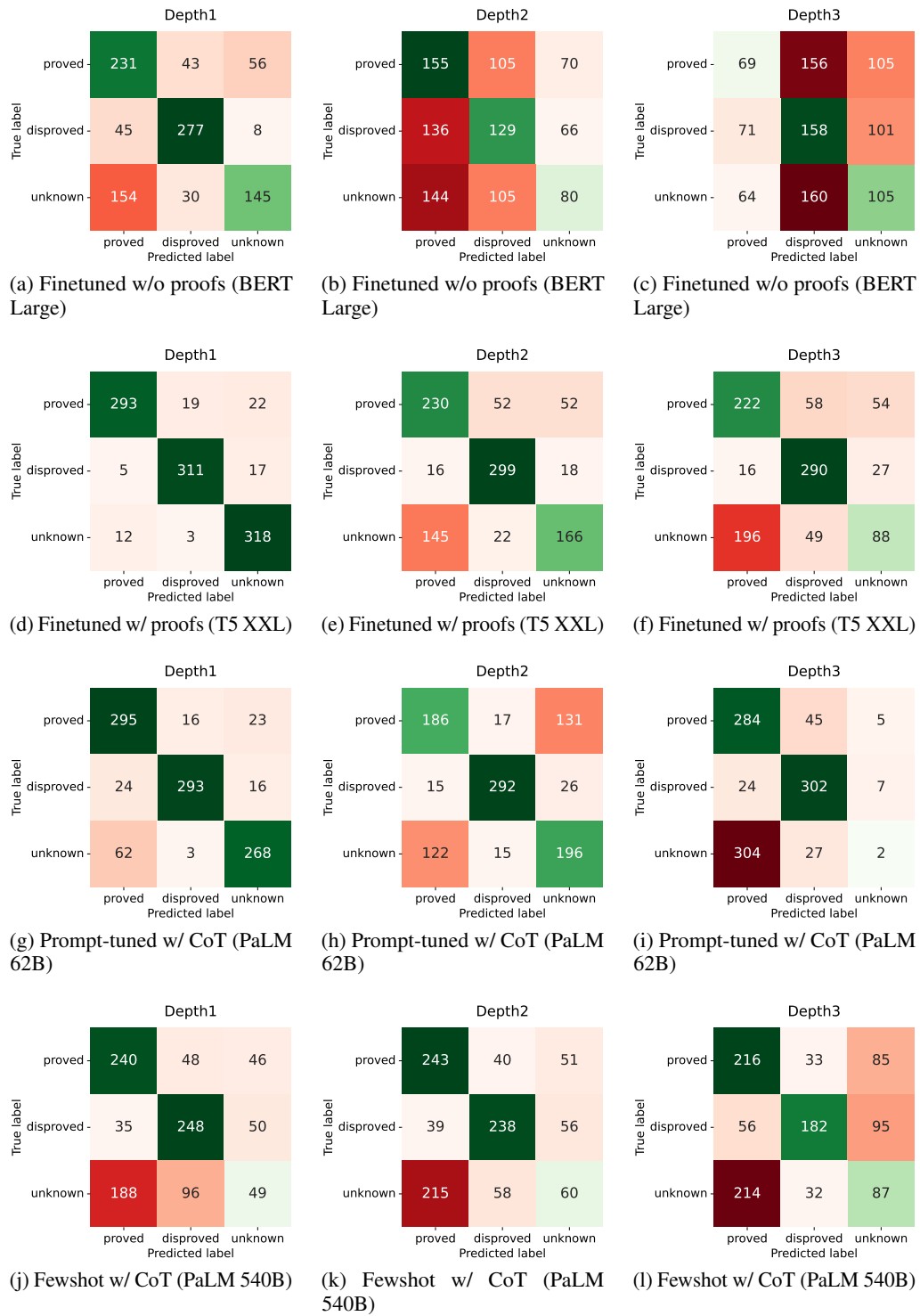

Figure 14: Confusion matrices for various models on the BoardgameQA dataset.

0.001 and allowed 50K epochs (since it required more time to converge) and reported the results for the epoch that achieved best validation accuracy.

> **Facts:** The seahorse has a basketball with a diameter of 27 inches, and is currently in Istanbul. The wolf does not take over the emperor of the snake. And the rules of the game are as follows.
>
> **Rules:** [...] Rule2: If the seahorse has a basketball that fits in a 37.1 x 37.4 x 23.9 inches box, then the seahorse surrenders to the swan. Rule3: The seahorse will surrender to the swan if it (the seahorse) is in Turkey at the moment. [...]
>
> **Preferences:** Rule5 is preferred over Rule4.
>
> **Question:** Does the swan swear to the woodpecker?
>
> **Prediction:** Since the seahorse is in Istanbul and according to Rule3 if the seahorse is in Turkey at the moment then the seahorse surrenders to the swan so the seahorse surrenders to the swan. Since the seahorse surrenders to the swan and according to Rule2 if the seahorse has a basketball that fits in a 37.1 x 37.4 x 23.9 inches box then the seahorse surrenders to the swan so the seahorse has a basketball that fits in a 37.1 x 37.4 x 23.9 inches box. [...]

Figure 15: An example of a wrong proof generated by PaLM 540B (fewshot) where the error is due to misunderstanding a logical rule (given a fact $f$ and a rule $f' \rightarrow f$ the model concludes that $f'$ must be true.

> **Facts:** The mule dances with the fish. [...]
>
> **Rules:** [...] Rule2: There exists an animal which dances with the fish? Then, the dachshund definitely does not capture the king (i.e. the most important piece) of the owl. Rule3: If something does not capture the king (i.e. the most important piece) of the owl and additionally not disarm the zebra, then it disarms the woodpecker. [...]
>
> **Preferences:** Rule1 is preferred over Rule5. Rule3 is preferred over Rule4.
>
> **Question:** Does the dachshund disarm the woodpecker?
>
> **Prediction:** [...] and according to Rule4 [...] and for the conflicting and higher priority rule Rule3 we cannot prove the antecedent "the dachshund does not capture the king of the owl", [...].

Figure 16: An example of a wrong proof generated by PaLM 62B (prompt-tuned) where the error is due to assuming an antecedent of a high priority rule cannot be proved, whereas it can indeed be proved.

## C  BoardgameQA Details

Here, we provide more in depth details about the generation and properties of the BoardgameQA dataset. A sample of theory and question generation from Algorithm 1 is provided in Figure 25.

### C.1  Consistency of the Dataset

A defeasible theory is called *consistent* if whenever a conflict arises, the preferences can be used to resolve the conflict. In BoardgameQA, we aim to produce consistent theories. To avoid inconsistencies and loops, each time we call the function in Algorithm 1, we only allow it to sample from the entities that have not been used in other rules and (sub-)questions. As an example, if we have a rule such as $a \wedge b \Rightarrow c$, then when we call the algorithm for $a$ and $b$ recursively, we use separate entities in the facts and rules produced for the sub-branch of $a$ and for the sub-branch of $b$. This way, we ensure that when we derive new facts in the sub-branch of $a$, it does not defeat some of the derivations in the sub-branch of $b$ (and the derivations in the later stages of the proof do not defeat the earlier derivations). That is because the the set of facts used in the rule bodies are for separate entities, and are therefore disjoint. We also apply defeasible reasoning on the final logical theory to ensure that the question can be derived from the theory.

### C.2  Incomplete Information

Here, we provide more information about the nature and type of incomplete information in BoardgameQA.

- **Age:** We first generate a positive integer $x$ corresponding to the age of one of the players expressed in days. Then we decide if we want to use a *more than* or *less than* relationship. In the case of the former, we next generate another integer $y < x$ and in the case of the latter $y > x$. Then, we randomly decide a target unit (days, weeks, months, or years) for each integer and convert them to that unit. Let $x'$ and $y'$ represent the obtained values measured with the new units. Then we add a fact *[PLAYER] is $x'$ [unit] old* and a rule *if the [PLAYER] is [more than/less than] $y'$ [unit] old then ...*. The model has to be able to convert units of time and then compare them.

> **Facts:** The mule dances with the fish. [...]
>
> **Rules:** [...] Rule2: There exists an animal which dances with the fish? Then, the dachshund definitely does not capture the king (i.e. the most important piece) of the owl. Rule3: If something does not capture the king (i.e. the most important piece) of the owl and additionally not disarm the zebra, then it disarms the woodpecker. [...]
>
> **Preferences:** Rule1 is preferred over Rule5. Rule3 is preferred over Rule4.
>
> **Question:** Does the dachshund disarm the woodpecker?
>
> **Prediction:** [...] and according to Rule4 [...] and for the conflicting and higher priority rule Rule3 we cannot prove the antecedent "the dachshund does not capture the king of the owl", [...].

Figure 17: An example of a wrong proof generated by PaLM 62B (prompt-tuned) where the error is due to filling the missing information incorrectly.

> **Facts:** The cougar manages to convince the ostrich, and smiles at the beetle. The dalmatian is named Chickpea. The pelikan has 8 friends. The seal is named Charlie.
>
> **Rules:** Rule1: If something smiles at the beetle and manages to convince the ostrich, then it negotiates a deal with the badger. Rule2: If the dalmatian has a name whose first letter is the same as the first letter of the seal's name, then the dalmatian invests in the company whose owner is the beetle. [...] Rule4: From observing that an animal brings an oil tank for the shark, one can conclude the following: that animal does not shout at the badger. Rule5: If the cougar negotiates a deal with the badger and the pelikan shouts at the badger, then the badger falls on a square of the mouse.
>
> **Preferences:** Rule4 is preferred over Rule3.
>
> **Question:** Does the badger fall on a square of the mouse?
>
> **Prediction:** Since the cougar manages to convince the ostrich and smiles at the beetle and according to Rule1 [...] so the cougar negotiates a deal with the badger. Since the cougar negotiates a deal with the badger and according to Rule5 if the cougar negotiates a deal with the badger and the pelikan shouts at the badger then the badger falls on a square of the mouse and Rule5 is preferred over the conflicting and lower priority rule Rule4 so the badger falls on a square of the mouse. So the statement the badger falls on a square of the mouse is proved and the answer is "yes".

Figure 18: An example of a wrong proof (but correct label) generated by PaLM 540B (fewshot) where the error is due to 1- failing to prove one element of the conjunction and also identifying a non-existence conflict between two rules.

- **Affordance:** We manually wrote object properties/affordances, and a list of items that have those properties. Examples of properties include *sharp*, *drink*, and *music* and examples of objects for each of these properties include *knife*, *cappuccino*, and *flute* respectively. The facts are of the form *The [PLAYER] has a [OBJECT]* and the rules are of the form *If the [PLAYER] has [AN OBJECT WITH PROPERTY] then ...*. The model has to know about the object properties to connect the facts and the rules.

- **Colors:** We first identify groups of colors based on some property. This includes *being a primary color*, *being a rainbow color*, and *being a color in the flag of country X*. Then, we generate facts of the form *The [PLAYER] has a card which is [COLOR] in color.* and rules of the form *If the [PLAYER] has a card whose color is [COLOR PROPERTY] then ...*. The model has to have information about colors to connect the facts and the rules.

- **Money:** We first generate a positive integer $x$ corresponding to the amount of money a player has, then we randomly decide if we want the comparison to be between two or three players. We also decide if we want to use a *more than* or *less than* relation. If the comparison is between two players and *more than* is used, then we generate another integer $y < x$ and if less than is used then $y > x$; if the comparison is between three players and *more than* is used then we generate $y, z$ such that $y + z < x$, and if *less than* is used then $y + z > x$. We then generate facts of the form *The [PLAYER i] has x dollars.* and rules of the form *If [PLAYER i] has [more than/less than] than [PLAYER j] and [PLAYER k] combined, then ...*. The model has to do a summation and decide which quantity is more or less.

- **Textual Entailment:** We manually write multiple pairs of sentences where one implies the other. Examples include *(assassinated the mayor, killed the mayor)*, *(struggles to find food, has difficulty to find food)*, and *(purchased a luxury aircraft, owns a luxury aircraft)*. The first element of the pair is used in the fact and the second element in the body of a rule. The model has to identify the entailment.

- **Places:** We manually write a list of cities and the countries they are located in. The city names are used in the facts (*The [PLAYER] is in [CITY] right now.*) and the countries are used in the rule bodies (*If the [PLAYER] is in [COUNTRY] right now, then ...*. The model has to know which city is in which country.

**Facts:** The monkey has a card that is orange in color. The monkey has a football with a radius of 29 inches, and is a farm worker.

Figure 19: An example of a wrong proof generated by PaLM 540B (fewshot) where the error is due to 1- mistaking a football with a basketball, and 2- not being able to fill in the incomplete information by realizing that a ball with a radius of 29 inches does not fit in a 26.3 x 25.6 x 24.2 inches box.

Figure 20: An example of a wrong proof generated by PaLM 540B (fewshot) where the error is due to starting with a distracting fact that took the proof on a wrong path (the correct proof is to first use the fact *The woodpecker swears to the duck* and Rule7 to conclude that *The woodpecker leaves the houses occupied by the dragon*, and then use Rule5 to conclude that *The vampire does not disarm the bulldog*.

- **Names:** We assign a name (from a list of manually written names) to two of the players and then write rules in the form of *If [PLAYER i] has a name that starts with the same letter as [PLAYER 2], then ... .*

- **Jobs:** We manually write a list of pairs of jobs and the industry they belong to. Examples include *(nurse, healthcare), (high school teacher, education),* and *(sales manager, marketing).* We use the job in the fact and the industry in the rule body. The model has to know which job is part of which industry.

- **Volume:** The facts mention that one of the players has an object (a notebook or a ball) and give the dimensions of the object (the height and width for notebook and the radius or diameter for the ball). The rule body asks whether the object fit in a box with some given dimensions. The model has to understand how 3D objects fit inside each other to be able to connect the fact to the rule.

- **Events:** We manually write a list of world events and the year when they occurred. Examples include *(world war 1 started, 1914), (the first man landed on moon, 1969)* and *(Obama's presidency started, 2009).* Then we write facts of the form *The [PLAYER] is watching a movie from [YEAR]* and rules of the form *If the [PLAYER] is watching a movie that was released [before/after] [EVENT], then ....* The model has to know the time for major world events to be able to connect the fact and the rule.

- **Friends:** We first generate a positive integer $x$ corresponding to the number of friends a player has. Then, we either generate a fact such as *The [PLAYER] has $x$ friends* or *The [PLAYER] has $x_1$ friends that are [ADJECTIVE] and $x_2$ that are not* where $x_1 + x_2 = x$. Then we decide if we want to use a *more than* or *less than* relation. In the former case, we generate a number $y < x$ and in the latter case $y > x$. Then we generate a rule of the form *If the [PLAYER] has [more than/less than] $y$ friends, then ....*

Due to the nature of the extra knowledge and reasoning cases we consider, we only add such cases at the last theory generation step of Algorithm 1 (i.e. when $d = 1$); Otherwise, we will need to follow

> **Facts:** The reindeer has three friends that are energetic and 1 friend that is not. The reindeer will turn 18 months old in a few minutes. The finch does not build a power plant near the green fields of the reindeer.
>
> **Rules:** Rule1: For the reindeer, if the belief is that the german shepherd disarms the reindeer and the finch builds a power plant near the green fields of the reindeer, then you can add that "the reindeer is not going to acquire a photograph of the finch" to your conclusions. Rule2: The leopard trades one of its pieces with the cobra whenever at least one animal builds a power plant close to the green fields of the finch. Rule3: Regarding the reindeer, if it is more than five years old, then we can conclude that it acquires a photo of the finch. Rule4: Here is an important piece of information about the reindeer: if it has fewer than 5 friends then it acquires a photo of the finch for sure.
>
> **Preferences:** Rule3 is preferred over Rule1. Rule4 is preferred over Rule1.
>
> **Question:** Does the leopard trade one of its pieces with the cobra?
>
> **Prediction:** Since [...] and according to Rule1 [...] and Rule1 is preferred over Rule3 so the reindeer is not going to acquire a photograph of the finch. [...]

Figure 21: An example of a wrong proof generated by PaLM 540B (fewshot) where the error is due to misunderstanding a preference.

> **Facts:** The liger has a card that is violet in color. The worm invests in the company whose owner is the mouse. The goat does not create one castle for the pigeon.
>
> **Rules:** Rule3: The living creature that does not create a castle for the pigeon will never hug the liger. [...]
>
> **Preferences:** Rule2 is preferred over Rule1. Rule6 is preferred over Rule5.
>
> **Question:** Does the liger invest in the company whose owner is the mermaid?
>
> **Prediction:** [...] We know the goat does not hug the liger, and according to Rule3 "if the goat does not hug the liger, then the liger does not hug the liger"

Figure 22: An example of a wrong proof generated by T5 where the error is due to hallucinating facts and rules.

generating sub-questions and rules for questions such as *The dog is named Paco* leading to unnatural rules such as *"If ... then the dog is named Paco*.

## C.3 Entities, Predicates, and Templates

Table 3 presents the set of entities, predicates, and templates used in BoardgameQA. To make the problem slightly more challenging in terms of language complexity, we use different entities, predicates and templates in the test set.

## C.4 Sample Proofs

In Figure 26, we provide some sample proofs with depths 1, 2, and 3 from the dataset. It can be observed that as the depth of the dataset increases, the length and difficulty of the proof increases substantially.

**Facts:** The llama creates one castle for the camel. The llama has a card that is white in color. The monkey is a public relations specialist.

**Rules:** [...] Rule2: Regarding the llama, if it has a card whose color appears in the flag of Italy, then we can conclude that it does not suspect the truthfulness of the mermaid. Rule3: If the monkey manages to convince the mermaid, then the mermaid is not going to pay money to the seahorse. Rule4: For the mermaid, if the belief is that the dinosaur dances with the mermaid and the llama does not suspect the truthfulness of the mermaid, then you can add "the mermaid pays some $$$ to the seahorse" to your conclusions. Rule5: If the monkey works in marketing, then the monkey manages to convince the mermaid.

**Preferences:** Rule1 is preferred over Rule2. Rule4 is preferred over Rule3.

**Question:** Does the mermaid pay money to the seahorse?

**Prediction:** We know the llama has a card that is white in color, white appears in the flag of Italy, and according to Rule2 [...]

Figure 23: An example of a wrong proof generated by T5 where the the model got distracted and ended up on a wrong proof path.

**Facts:** The otter has a 10 x 18 inches notebook. The otter is 15 months old. The duck does not tear down the castle that belongs to the otter.

**Rules:** Rule1: If the otter does not enjoy the companionship of the dugong, then the dugong captures the king (i.e. the most important piece) of the bear. Rule2: Regarding the otter, if it is more than four years old, then we can conclude that it does not want to see the dugong. Rule3: Regarding the otter, if it has a notebook that fits in a 18.1 x 12.8 inches box, then we can conclude that it does not want to see the dugong.

**Preferences:** None

**Question:** Does the dugong capture the king of the bear?

**Prediction:** Since the otter is 15 months old and 15 months is less than 4 years and according to Rule2 if the otter is less than 4 years old then it does not want to see the dugong so the otter does not want to see the dugong. [...]

Figure 24: An example of a wrong proof generated by PaLM 540B (fewshot) where the error is due to changing a rule in such a way that a proof can be found, when a proof does not exist.

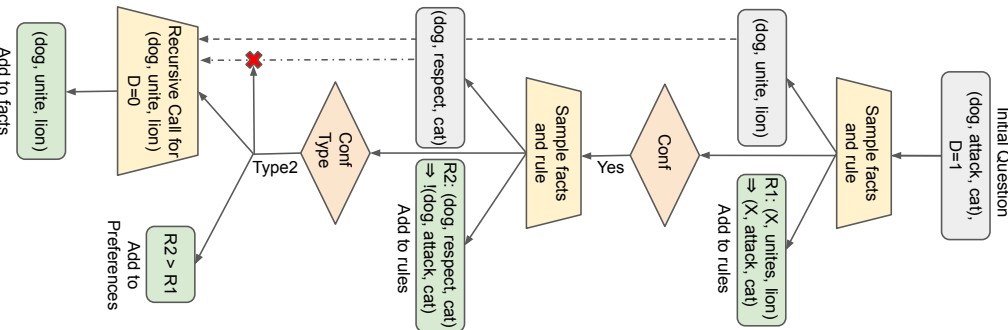

Figure 25: A sample of theory and question generation from Algorithm 1. Initially, the question has been selected to be *(dog, attack, cat)*. The input depth $D = 1$ indicates that a theory with one hop of reasoning should be generated. Then a fact *(dog, unite, lion)* and a rule *R1: (X, unite, lion)* $\Rightarrow$ *(X, attack, cat)* have been generated. Notice that the combination of the fact and the rule conclude *(dog, attack, cat)*. Next we randomly decide if a conflict should be generated. The decision is yes, so we generate another fact *(dog, respect, cat)* and rule *(dog, respect, cat)* $\Rightarrow$ *!(dog, attack, cat)*. Notice that the two rules have contradictory conclusions now. We next decide randomly the type of the conflict, and Type2 is selected in this case. Therefore, we add *R2 > R1* to our preferences, remove one of the facts generated for the conflicting rule, and make recursive calls for the remaining facts, which is only *(dog, unite, lion)*. This call is made with $D = 0$, therefore the stopping criterion triggers and we add *(dog, unite, lion)* to our set of facts.

**Sample Depth 1 Proof:** We know the dog has a card that is white in color, white starts with "w", and according to Rule2 "if the dog has a card whose color starts with the letter "w", then the dog prepares armor for the eel", and Rule2 has a higher preference than the conflicting rules (Rule1), so we can conclude "the dog prepares armor for the eel". So the statement "the dog prepares armor for the eel" is proved and the answer is "yes".

**Sample Depth 2 Proof:** We know the whale eats the food of the koala, and according to Rule2 "if something eats the food of the koala, then it rolls the dice for the leopard", so we can conclude "the whale rolls the dice for the leopard". We know the buffalo does not know the defensive plans of the cat, and according to Rule1 "if something does not know the defensive plans of the cat, then it doesn't attack the green fields whose owner is the leopard", so we can conclude "the buffalo does not attack the green fields whose owner is the leopard". We know the buffalo does not attack the green fields whose owner is the leopard and the whale rolls the dice for the leopard, and according to Rule4 "if the buffalo does not attack the green fields whose owner is the leopard but the whale rolls the dice for the leopard, then the leopard raises a peace flag for the zander", and for the conflicting and higher priority rule Rule3 we cannot prove the antecedent "the leopard does not respect the kudu", so we can conclude "the leopard raises a peace flag for the zander". So the statement "the leopard raises a peace flag for the zander" is proved and the answer is "yes".

**Sample Depth 3 Proof:** We know the crocodile is named Milo and the donkey is named Meadow, both names start with "M", and according to Rule11 "if the crocodile has a name whose first letter is the same as the first letter of the donkey's name, then the crocodile steals five points from the cockroach", and for the conflicting and higher priority rule Rule4 we cannot prove the antecedent "at least one animal knows the defensive plans of the buffalo", so we can conclude "the crocodile steals five points from the cockroach". We know the crocodile steals five points from the cockroach, and according to Rule6 "if something steals five points from the cockroach, then it does not remove from the board one of the pieces of the tiger", so we can conclude "the crocodile does not remove from the board one of the pieces of the tiger". We know the aardvark proceeds to the spot right after the lion, and according to Rule8 "if something proceeds to the spot right after the lion, then it does not eat the food of the spider", and for the conflicting and higher priority rule Rule3 we cannot prove the antecedent "at least one animal attacks the green fields whose owner is the catfish", so we can conclude "the aardvark does not eat the food of the spider". We know the aardvark does not hold the same number of points as the phoenix, and according to Rule7 "if something does not hold the same number of points as the phoenix, then it doesn't give a magnifier to the eagle", so we can conclude "the aardvark does not give a magnifier to the eagle". We know the aardvark does not give a magnifier to the eagle and the aardvark does not eat the food of the spider, and according to Rule2 "if something does not give a magnifier to the eagle and does not eat the food of the spider, then it knows the defensive plans of the tiger", so we can conclude "the aardvark knows the defensive plans of the tiger". We know the aardvark knows the defensive plans of the tiger and the crocodile does not remove from the board one of the pieces of the tiger, and according to Rule10 "if the aardvark knows the defensive plans of the tiger but the crocodile does not remove from the board one of the pieces of the tiger, then the tiger becomes an enemy of the cheetah", so we can conclude "the tiger becomes an enemy of the cheetah". So the statement "the tiger becomes an enemy of the cheetah" is proved and the answer is "yes".

Figure 26: Sample proofs from the dataset at depths 1, 2 and 3. Higher depth proofs are substantially longer than the lower depth proofs.

| | |
|---|---|
| Train entities | cat, dog, pig, parrot, eagle, squirrel, penguin, lion, tiger, donkey, leopard, cheetah, grizzly bear, polar bear, sun bear, panda bear, black bear, turtle, crocodile, elephant, panther, cow, rabbit, hare, buffalo, baboon, sheep, whale, jellyfish, carp, goldfish, viperfish, starfish, catfish, oscar, zander, sea bass, swordfish, salmon, halibut, blobfish, doctorfish, tilapia, kangaroo, octopus, phoenix, aardvark, amberjack, eel, hummingbird, canary, hippopotamus, snail, caterpillar, mosquito, bat, ferret, gecko, kudu, moose, cockroach, cricket, grasshopper, meerkat, spider, lobster, squid, puffin, raven, kiwi, koala, wolverine |
| Test entities | akita, bear, camel, coyote, snake, monkey, leopard, fish, ostrich, pigeon, dolphin, frog, goat, goose, wolf, gorilla, beaver, lizard, flamingo, swan, elk, duck, reindeer, bison, shark, mouse, owl, llama, cobra, zebra, otter, crab, peafowl, rhino, dinosaur, dove, badger, chinchilla, cougar, crow, seal, worm, ant, bee, butterfly, dragonfly, dragon, gadwall, mule, liger, german shepherd, bulldog, husky, poodle, chihuahua, dachshund, basenji, dalmatian, mermaid, seahorse, fangtooth, dugong, walrus, vampire, stork, swallow, songbird, woodpecker, starling, mannikin, pelikan, beetle, finch |
| Train predicates | owe money to, give a magnifier to, learn the basics of resource management from, know the defensive plans of, show all her cards to, prepare armor for, sing a victory song for, need support from, respect, raise a peace flag for, become, an enemy of, roll the dice for, hold the same number of points as, offer a job to, wink at, steal five points from, knock down the fortress of, burn the warehouse of, eat the food of, attack the green fields whose owner is, proceed to the spot that is right after the spot of, remove one of the pieces of |
| Test predicates | tear down the castle that belongs to, bring an oil tank for, reveal a secret to, enjoy the company of, neglect, want to see, swear to, refuse to help, manage to convince, call, stop the victory of, dance with, shout at, smile at, pay money to, unite with, hug, destroy the wall constructed by, create one castle for, disarm, acquire a photograph of, borrow one of the weapons of, fall on a square of, suspect the truthfulness of, invest in the company whose owner is, leave the houses occupied by, hide the cards that she has from, swim in the pool next to the house of, negotiate a deal with, trade one of its pieces with, build a power plant near the green fields of, take over the emperor of, capture the king of, surrender to |
| Train templates (sampled) | **Universally quantified rule:** If something [A] the [B], then it does not [C] the [D]. 
 **Existentially quantified rule:** If at least one animal [A] the [B], then the [C] [D] the [E]. 
 **No Quantifier:** For the [A], if the belief is that the [B] does not [C] the [A] and the [D] does not [E] the [A], then you can add "the [A] does not [F] the [G]" to your conclusions. |
| Test templates (sampled) | **No Quantifier:** In order to conclude that the [A] does not [B] the [C], two pieces of evidence are required: firstly that the [D] will not [E] the [A] and secondly the [F] [G] the [A].' 
 **Existential Quantifier:** There exists an animal which [A] the [B]? Then, the [C] definitely does not [D] the [F]. 
 **Universal Quantifier:** From observing that one animal [A] the [B], one can conclude that it also [C] the [D], undoubtedly. |

Table 3: Categories, descriptions, and examples of incomplete information in BoardgameQA. For lexical entailment, world knowledge, event times, and affordance, a list of examples is written manually from which the sampling procedure can select. In the other cases, examples are generated automatically.

