# OpenReview forum: "BoardgameQA: A Dataset for Natural Language Reasoning with Contradictory Information"
_NeurIPS.cc/2023/Track/Datasets_and_Benchmarks — NeurIPS 2023 Datasets and Benchmarks Poster_

### Official Review · Reviewer_MReC · 2023-07-20
**This work presents an interesting dataset for natural language reasoning, analyzes existing LMs, and discusses their limitations in exhibiting reasoning abilities.**

**Rating:** 6
**Confidence:** 4

**Strengths:**

- It is important to analyze the various reasoning abilities of existing and widely used LMs. Here, the authors proposed a dataset to measure the reasoning ability with conflicting sources guided by preferences over sources (defeasible reasoning).

- This paper proposed an algorithm to generate dataset in an effective way for different depths of reasoning.

- In this work, the authors conducted a thorough analysis of different types of models in various experimental setups on BoardgameQA and presented several interesting findings about models’ defeasible reasoning ability. The findings suggest that existing LMs do not fare well for preference understanding and defeasible reasoning indicating the scope for developing more robust models for reasoning over contradicting and incomplete sets of information.

**Additional Feedback:**

It is advisable to have GitHub or another platform for the dataset access along with documentation and license to use.

**Clarity:**

The paper is well written and all notations are explained properly. However, there is a detailed explanation is required for the “Proofs” in Section 4 since I am not able to understand clearly what the proofs are from the explanation.

**Correctness:**

The findings of this work are interesting and the experiments are thorough. The dataset is created using an effective algorithm. However, see the suggestions and questions in the above “Opportunities For Improvement” for more details.

**Documentation:**

Yes, there is sufficient detail on data collection and organization, availability and maintenance, and ethical and responsible use.


**Limitations:**

Yes, the authors adequately addressed the limitations and potential negative societal impact of their work.

**Opportunities For Improvement:**

Related to Data Generation:

1. Authors state that they adopt a set of 6 rule templates containing existential and universal quantifiers, conjunctions, and missing information. However, I am not sure about the significance of these quantifiers in the data generation. There is no specific detailed discussion on the utilization of these quantifiers. Authors need to carefully explain this and provide examples of how they are used in natural language.

2. In paragraphs “Selecting a question” and “Conversion to natural language”, it is mentioned that the templates are used to create questions and also to convert facts, rules, and preferences into natural language. Why not use the existing LLMs generation ability to generate more diverse and natural samples?

3. Section 4 indicates that disproved are created simply by negating the question, however, creating negations for the logical sentences is not an easy task, especially when they involve inference rules such as modus ponens and quantifiers. The process for generating negation is not explained clearly in the paper. How are the negations created? Is there any specific algorithm used to take care of logical rules and connectives? It is important to know this process.

Related to Experiments:

1. Authors have covered a range of models, however, experimenting with the GPT-family models such as GPT-4, ChatGPT, and GPT-3 also can add value to this work in terms of supporting the analysis. Furthermore, it is easier to analyze freely available models such as GPT-Neo, or GPT-J.

2. Authors should present the statistical significance of the results for the freely available models as BERT and T5 (results over multiple seeds).

3. Presenting an analysis with a few samples where the depth of the analysis is incrementally increased from 1 to n, where n>3. This demonstration aims to showcase the escalating difficulty of examples when surpassing a depth of 3. Such an investigation promises to be an intriguing and informative analysis.

4. Authors claim that the BoardgameQA can help in measuring the natural language reasoning ability, however, the dataset is generated using a template-based method which inherently restricts the extent of variations achievable in terms of natural language expressions. How authors have incorporated natural language variations? Having this explanation is important.

**Relation To Prior Work:**

Yes, it clearly discussed how this work differs from previous contributions.


**Summary And Contributions:**

In this paper, authors present BoardgameQA, a synthetic dataset for measuring the multi-hop defeasible reasoning ability (reasoning with conflicting sources by preference) of existing LMs. In this dataset, they propose an algorithm to generate facts, rules, preferences, and questions; and template-based natural language conversion to turn each of them into a textual format. Authors evaluate LMs such as BERT (encoder only), T5 (encoder-decoder), and PaLM (decoder only) models on BoardgameQA on depths 1-3. Furthermore, they present a detailed analysis of models on depth 2 (medium level of difficulty) and present interesting findings.

---

> ### Author Response · Authors · 2023-08-15
> **Author Response**
>
> > Presenting an analysis with a few samples where the depth of the analysis is incrementally increased from 1 to n.
>
> We added Figures 11, 12, and 13 in the appendix which showcase some examples of errors for depth 1, depth 2, and depth 3 examples respectively. Moreover, we added Figure 26 in the appendix which shows sample proofs from the dataset with depths 1, 2, and 3 to showcase how the solution becomes substantially more difficult as the depth of the proof increases. We are also planning on generating subsets of our dataset with higher depths and release them publicly for further future analysis and investigation.
>
> > How authors have incorporated natural language variations?
>
> Natural language variations have been incorporated in the following ways:
> * For each rule and for each assignment of predicate signs, we write multiple templates (100 templates overall). Some examples are provided in Table 3 of the supplementary material. A subset of these templates are only used in the test set to ensure the model does not overfit to the provided templates.
> * We have a large set of entities and predicates (see Table 3 of the supplementary material). Unlike existing work that uses single word predicates (e.g., chase, like, etc.), we use long, multi-word predicates. The set of entities and predicates in the train and test sets are mutually exclusive to ensure the models do not overfit to the training entities and predicates.
> * For each predicate, we write it in two ways and each time we want to use it, we randomly select one of the variations. Examples include: 1- “destroy the wall built by” and “destroy the wall constructed by” and 2- “fall on a square of” and “fall on a square that belongs to”.
> * We include several variations of specifying the facts. Examples include: 1- writing numbers as 42 or forty two, and 2- sometimes combining two facts about an entity into one fact, e.g. combining “The dog is in Montreal. The dog has a knife.” into “The dog is in Montreal and has a knife.”.
>
> > There is no specific detailed discussion on the utilization of these quantifiers.
>
> Examples of how quantification is used in natural language is provided in Table 3 of the supplementary material (see the last two rows). Abstractly, a rule with no quantifier such as `p(a, b) => q(c, d)` translates to `if a p the b then c q the d`, a rule with universal quantifier such as `\forall X: p(X, a) => q(X, b)` translates to `if someone p the a, then they will q the b`, and a rule with existential quantifier such as `\exists X: p(X, a) => q(b, c) translates to `If someone p the a, then b will q the c`.
>
> > Why not use LLMs instead of templates?
>
> We decided to use templates instead of LLMs for two reasons:
> * For our experiments, we needed to create multiple large variants of our dataset. If we used LLMs to turn theories into textual format, we needed to do human verification to ensure the generated text is semantically equivalent to the logical theory. Such a verification is costly for the size of our dataset, and the size of our examples/proofs.
> * If we use LLMs to generate the text of the questions and then measure the performance of the LLMs on those questions, the problems might become artificially simpler because different LLMs may have similar language biases.
>
> > creating negations for the logical sentences is not an easy task
>
> We clarify that we only negate the question, not the logical formulae. Suppose that we have created the following logical theory and question:
>
> **Facts**: The dog has 3 dollars. The lion has 8 dollars.
>
> **Rules**: If the dog has less money than the lion, it will not attack the lion.
>
> **Question**: The dog does not attack the lion.
>
> The original answer to the question is “Proved”. To create an example whose answer is “Disproved”, we use the same set of facts and rules, but change the question to:
>
> **New Question**: The dog attacks the lion.
>
> > Results over multiple seeds
>
> We added the standard deviations for the BERT model (which is much cheaper than the other models) in Appendix A.

---

### Official Review · Reviewer_ayn7 · 2023-07-21
**Contradictory Information Reasoning Dataset Review**

**Rating:** 7
**Confidence:** 3
**Clarity:** The paper is well-written.

**Strengths:**

1. The BoardgameQA dataset is of high quality and might be meaningful for the research in reasoning with contradictory information.

2. The experiments are comprehensive, and the results lead to several interesting findings.




**Additional Feedback:**

None

**Correctness:**

The paper overall seems correct, with a clear explanation of the dataset construction and the evaluation methods.

**Documentation:**

The paper contains details on how the data was collected and where it is maintained.

**Ethics:**

There seem to be no ethical concerns.

**Limitations:**

The authors discussed the limitations in the appendix.

**Opportunities For Improvement:**

1. In the dataset the entities such as dog, frog, and lion have their own semantic meaning, which may influence the experiment results. Experiments like randomly shuffling the entities will make the results more convincing.

2. The performance of PaLM 540B seems to be much worse than T5 XXL and PaLM 62B. While larger models often have stronger reasoning abilities, are there any possible explanations for the results?

**Relation To Prior Work:**

The paper discusses related work well.

**Summary And Contributions:**

The paper curates a new dataset called BoardgameQA for reasoning with contradictory information guided by preferences over sources. The authors evaluate several LMs on the dataset and make some interesting discoveries.

Contributions:

1. Build a new dataset for reasoning with contradictory information guided by preferences.
2. Benchmark several LMs on the dataset and make some interesting discoveries.

---

> ### Author Response · Authors · 2023-08-15
> **Author Response**
>
> > In the dataset the entities such as dog, frog, and lion have their own semantic meaning, which may influence the experiment results. Experiments like randomly shuffling the entities will make the results more convincing.
>
> We would like to first clarify that 1- we have a large pool of entity names (see Table 3 in supplementary material) and for each example we select a small subset of them, so the entities are already quite varied and shuffled across examples, 2- we use different entity names during training and testing to ensure that any biases the models learn about entity names during training is not useful for the test set examples.
>
> We conducted one more experiment where for our base dataset (the one from Figure 4 – depth 2) we replaced the entity names with random letters (e.g., replacing “cat” with “MYNBI”, “dog” with “QPMZJ”, etc.) to ensure that the model does not have much prior belief about them. We then tested PaLM 540b and FLAN PaLM 540b on the resulting dataset to see if the model performance improves in this setting. We observed that for PaLM the performance went down from 54.1 to 50.3 and for FLAN PaLM went up from 48.4 to 49.9. This shows that the model belief about the entities does not have a substantial negative impact on its performance. This is in line with the observation in reference [41] of our submission.
>
> > The performance of PaLM 540B seems to be much worse than T5 XXL and PaLM 62B. While larger models often have stronger reasoning abilities, are there any possible explanations for the results?
>
> The PaLM 540B model is tested in a fewshot setting whereas the T5XXL and PaLM 62B models have been tested in a supervised finetuning and soft-prompt tuning setting respectively. That is why PaLM 540B underperforms compared to the other two models. However, in Figure 4 we also report results for fewshotting PaLM 62B and it can be observed that under that setting, PaLM 540B outperforms PaLM 62B substantially.

---

> > ### Comment · Reviewer_ayn7 · 2023-08-29
> >
> > Thank you for answering my questions and providing clarifications. I have no further questions.

---

### Official Review · Reviewer_Gau4 · 2023-07-21
**Well constructed and evaluated benchmark on reasoning with conflicts**

**Rating:** 7
**Confidence:** 3

**Strengths:**

- A new benchmark dataset emulating reasoning in real-world scenario with conflicting information and reasoning with implicit knowledge
- Broad empirical comparisons to show the challenging nature of this task
- Careful analysis to highlight the key challenges of this benchmark (contradictions, type of conflicts, etc.)


**Additional Feedback:**

- Is there a chance that there are alternate but valid proofs for certain questions, even if they were intended to be invalid?

- The model is expected to fill in the gaps when incomplete information is provided. But at the same time, it should not fill in the gaps when it comes to rules where one of the fact is deliberately dropped (#203). Doesn't that make the task specification a bit unclear? I understand that the facts in the context are very artificial but models may have certain beliefs about them; just like they have beliefs about facts like "Montreal is in Canada".

- Why the choice to subsample in line 10 in Algo 1 vs removing only one sub-question in line 13?


**Clarity:**

The paper is pretty well-written. It would help if Alg. 1 also incorporated the incompleteinformation example generation.


**Correctness:**

The dataset has been constructed synthetically and should be correct to the extend the code is correct. It would be valuable to have an oracle solver (e.g. defeasible solver in #244) score and ensure that it achieves 100% on the benchmark.


**Documentation:**

I am not sure about the dataset's hosting, licensing, and maintenance plan.


**Limitations:**

The paper does have a dedicated limitation section (but in the Appendix which I missed initially). It would be worth presenting these limitations in the main paper.


**Opportunities For Improvement:**

- It would be great if the facts also had associated source or time stamps where certain sources had a higher preference. While this can be emulated with preference on rules too, but it would be worth seeing if models are able to handle conflicts in facts better than conflicts via rules.

- For a synthetically generated task, max depth of 3 seems a bit small. While models do currently find even these tasks to be challenging, higher depth theories would make this benchmark more future proof.

- Other than simple baselines, it would be great if the authors compared to more structured reasoning approaches. E.g LAMBADA.


**Relation To Prior Work:**

Table 1 clearly compares the current benchmark to prior work. There were couple of papers in ACL'23 that might also be worth contrasting with (False Premises & False Presuppositions).


**Summary And Contributions:**

The paper presents a new benchmark on reasoning with conflicting information. The benchmark (BoardGameQA) contains QA problems with facts and rules that can lead to conflicting conclusions. The rules have associated preference order that should be used to resolve conflicts. This is meant to emulate real life scenarios where one may find conflicting information but can choose the more trustworthy source or newest piece of information to resolve this conflict. In addition, questions may also need background information that is not provided in the context to answer them. Experiments with fine-tuned models and prompted LLMs show that this is a challenging task for current models.

---

> ### Author Response · Authors · 2023-08-15
> **Author Response**
>
> > There were couple of papers in ACL'23 that might also be worth contrasting with (False Premises & False Presuppositions).
>
> Thanks for bringing them to our attention. Both datasets (CREPE and FalseQA are added to Table 1.)
>
> > It would be worth presenting these limitations in the main paper.
>
> Upon your suggestion, the limitations are now (condensed and) moved to the main body.
>
> > It would be valuable to have an oracle solver (e.g. defeasible solver in #244) score and ensure that it achieves 100% on the benchmark.
>
> We clarify that we have already ran the solver (described in #244) on all the examples.
>
> > It would help if Alg. 1 also incorporated the incomplete information example generation.
>
> Upon your suggestion, we expanded Alg 1 into Alg 1 and Alg 2 and described the procedure for incomplete information.
>
> > I am not sure about the dataset's hosting, licensing, and maintenance plan.
>
> The dataset will be hosted and maintained in Google cloud storage (GCS), under the Google Research bucket (https://storage.googleapis.com/gresearch/BoardgameQA/BoardgameQA.zip) with a CC BY license to allow broad usage.
>
> > For a synthetically generated task, max depth of 3 seems a bit small.
>
> Thanks for the suggestion. We stopped at depth 3 because we observed that current models struggle even at that depth. But we agree that having higher depths will make the dataset more future proof. We will generate datasets of higher depths and will release them publicly.
>
> > It would be great if the facts also had associated source or time stamps where certain sources had a higher preference.
>
> In our binary setting, conflicts over facts can be removed in a separate preprocessing step before feeding them into the reasoner. For example, if we have “Fact1: A is true, Fact2: A is False, Fact1 is preferred over Fact2”, then the preprocessing step can easily remove Fact2 before the example is fed into the reasoner. Note, however, that this is not true for the case of rules because if we have “Rule1: A->C, Rule2: B->!C, Rule1 is preferred over Rule2”, then the reasoner can still use Rule2 for the examples where B is true and A is False. For this reason, and because conflicts over facts are not very common in defeasible reasoning, we excluded them from the dataset.
>
> We note, however, that there could be some extensions of our setting with facts such as “Fact1: Alice has visited all countries in Europe, Fact2: Alice has not visited France, Fact2 is preferred over Fact1”, where the reasoner can use both facts. This is now mentioned in the limitations section.
>
> Also, note that facts can be thought of as special rules with an empty body.
>
> > Compare to LAMBADA
>
> Our goal in the current project was to benchmark LLMs in isolation and show the gap in their reasoning capabilities with contradictory information. Future work can indeed experiment with various techniques that either supplement LLMs with reasoning algorithms (e.g., extensions of LAMBADA to defeasible reasoning), LLMs with tool-use, or even develop custom strategies tailored to textual defeasible reasoning.
>
> > Is there a chance that there are alternate but valid proofs for certain questions, even if they were intended to be invalid?
>
> This does not happen when using Algorithm 1 to generate the theories. To understand why, one has to notice to things:
>
> 1- (Ignoring conjunctions for ease of explanation) Algorithm 1 starts from the question and generates a proof chain step by step in the backward direction. The only case where the chain turns into a tree (thus enabling the possibility of multiple proofs) is where we introduce conflicts. But in the case of conflicts, even if both branches of the tree are activated, our preferences guide us into selecting only one of those branches, therefore generating at most one valid proof.
>
> 2- We use different entities for our rules and facts in different branches and in different steps of the backward generation. Therefore, there can be no jumping rules/facts, where the facts generated for one branch accidentally activate one of the rules in another branch.

---

> > ### Author Response · Authors · 2023-08-15
> > **Author Response Continued**
> >
> > > models may have certain beliefs about artificial facts; just like they have beliefs about facts like "Montreal is in Canada".
> >
> > As humans, when we read the rules of a board game, we need to combine a large amount of our world knowledge with the artificial rules of the game and we can typically understand the distinction clearly. Therefore, we expect that a strong LLM should also be able to understand the distinction and reason accordingly. An evidence for this is the results of the PaLM 540B model in Figure 8 where the model shows similar performance when making the problem knowledge-heavy.
> >
> > We conducted one more experiment where for our base dataset (the one from Figure 4 – depth 2) we replaced the entity names with random letters (e.g., replacing “cat” with “MYNBI”, “dog” with “QPMZJ”, etc.) to ensure that the model does not have much prior belief about them. We then tested PaLM 540b and FLAN PaLM 540b on the resulting dataset to see if the model performance improves in this setting. We observed that for PaLM the performance went down from 54.1 to 50.3 and for FLAN PaLM went up from 48.4 to 49.9. This shows that the model belief about the entities does not have a substantial negative impact on its performance. This is in line with the observation in reference [41] of our submission.
> >
> >
> > > Why the choice to subsample in line 10 in Algo 1 vs removing only one sub-question in line 13?
> >
> > The choice to subsample in line 10 of the algorithm is primarily motivated by creating problems at different levels of difficulty/width. Note that in subsampling, each sub-question is removed with some probability independent of the other sub-questions, so it is possible that in some cases none of the sub-questions are removed. In line 13, however, one of the sub-questions has to be removed to ensure that the rule does not activate, which is why we remove one of the sub-questions instead of sub-sampling.

---

### Official Review · Reviewer_cpZF · 2023-07-22
**Review of paper "BoardgameQA: A Dataset for Natural Language Reasoning with Contradictory Information"**

**Rating:** 7
**Confidence:** 3
**Clarity:** The paper is really well written.

**Strengths:**

The current paper nicely covers all the required explanations and information for a work of this kind. Therefore, the main strengths I could highlight are:

- The paper tackles an important task for current LMs, devising a dataset that outperforms previous ones.

- The authors provide a detailed explanation of related work, focusing specifically on previous datasets with the same purpose, and their caveats.

- The creation of the dataset is adequately explained and, at least for me, stresses further the validity and interest of the data presented in the current paper.

- Beyond, the dataset is quite rich, in terms of the criteria considered for its creation, the different levels of complexity, the different types of incomplete information, etc.

- The list of experiments is exhaustive, tackling many different questions of interest for the understanding of how LMs operate to solve the task of automated reasoning with incomplete information.

For all these reasons, I believe the paper is quite strong, and the dataset presented can provide a lot of value to the NLP community, and help improve LMs in such challenging and relevant task.


**Additional Feedback:**

I could only find a small typo: in all figures, Finetuned is misspelled as Fineutned.

**Correctness:**

I believe all is nicely justified, and the explanation of the dataset creation is detailed and intuitive. Thanks to the experiments presented, it is possible to answer the most important questions about the validity of the dataset, and how current LMs behave when tackling this task.

**Documentation:**

Yes, there is. The authors provide a link to the dataset and specify the license for its usage.

**Ethics:**

No, there are not.

**Limitations:**

As stated before, I don’t think it is possible to highlight any limitations. Different ways of expanding further the dataset are also discussed in the paper as future work.

**Opportunities For Improvement:**

Honestly, I don’t think the presented dataset requires any further enhancement. Obviously, it could be expanded with more rules, a reasoning task that does not only requires a category as output, etc. But all these limitations are already listed and discussed by the authors, which also see them as potential avenues for improving the dataset. But in its current state, it is already a really meritorious dataset with a lot of potential.

**Relation To Prior Work:**

I do appreciate a lot that the authors have focused in the Related Work section on previous datasets used for benchmarking, concisely analyzing them and their main lacks, which they try to overcome with the proposed dataset.

**Summary And Contributions:**

In the current paper, the authors present a large and varied dataset to benchmark LMs in the task of automated reasoning in situations where there is contradictory, incomplete, or inconsistent information. In these cases, the LM is instructed to perform several hops of reasoning, in order to infer some intermediate answers, resolve missing information, etc., and decide between different contradictory options, according to some established rules. All these boundary conditions and complexities further complicate substantially the correct performance of current methods, as shown by the authors in the different tests carried out. In order to generate the dataset, the authors pose the task as a problem of defeasible reasoning, where a set of input facts, contradictory rules, and preferences are required, and generate all these elements with a backward story generation strategy. The dataset presented is quite complete, with different levels of complexity (required reasoning depth), a plethora of ways of incorporating incomplete information, and two types of possible confusion. Besides, all the tests performed by the authors touch upon all the important questions about the task under study, and how LMs may try to tackle it.

---

> ### Author Response · Authors · 2023-08-15
> **Author Response**
>
> We thank the reviewer for their encouraging words. And thanks for catching the typo in our figures; it is now fixed.

---

### Decision · Program_Chairs · 2023-09-22

**Decision:**

Accept (Poster)

**Comment:**

This paper presents BoardgameQA, a new benchmark dataset designed to evaluate LLM’s automated reasoning capabilities when there is conflicting, incomplete, or inconsistent information, i.e. QA problems contain facts and rules that can lead to conflicting conclusions. The experiments show that out-of-the-box LLMs do not handle these situations very well and finetuning does not seem to resolve the problem much, albeit small improvement. All the reviewers seem to agree that this dataset is a positive addition to the NLP community and all the reviewers hold a consensus that experiments and analysis are thorough.

There were comments that could further improve the paper. For example,
- max-depth 3 is too small for a synthetic dataset,
- conflicts in facts vs. conflicts via rules,
- comparison to the related work,
...

I believe the authors are on the way to generating max-depth>3 and already compared to more related work in Table 1. I recommend the authors to resolve/reflect comments like this in the camera ready.